# Using Data Mining and Network Analysis to Infer Arboviral Dynamics: The Case of Mosquito-Borne Flaviviruses Reported in Mexico

**DOI:** 10.3390/insects12050398

**Published:** 2021-04-29

**Authors:** Jesús Sotomayor-Bonilla, Enrique Del Callejo-Canal, Constantino González-Salazar, Gerardo Suzán, Christopher R. Stephens

**Affiliations:** 1Laboratorio de Ecología de Enfermedades y Una Salud, Departamento de Etología, Fauna Silvestre y Animales de Laboratorio, Facultad de Medicina Veterinaria y Zootecnia, Universidad Nacional Autónoma de México, Coyoacán, Ciudad de México 04510, Mexico; chuchomayor16@gmail.com; 2Asociación Mexicana de Medicina de la Conservación Kalaan kab AC, Coyoacán, Ciudad de México 04510, Mexico; 3Centro de Ciencias de la Complejidad, Universidad Nacional Autónoma de México, Coyoacán, Ciudad de México 04510, Mexico; edelcallejoc@gmail.com; 4Posgrado en Ciencia e Ingeniería de la Computación, Instituto de Investigaciones en Matemáticas y en Sistemas, Universidad Nacional Autónoma de México, Coyoacán, Ciudad de México 04510, Mexico; 5Departamento de Ciencias Atmosféricas, Centro de Ciencias de la Atmósfera, Universidad Nacional Autónoma de México, Coyoacán, Ciudad de México 04510, Mexico; cgsalazar@atmosfera.unam.mx; 6Departamento de Gravitación y Teoría de Campos, Instituto de Ciencias Nucleares, Universidad Nacional Autónoma de México, Coyoacán, Ciudad de México 04510, Mexico

**Keywords:** vector-borne diseases, disease transmission cycles, vector-host system, multi-pathogen model, Dengue virus, Yellow fever virus, St. Louis encephalitis virus, West Nile virus, spatial distribution models, complex networks

## Abstract

**Simple Summary:**

Given the significant impact on both human and animal health of mosquito-borne flaviviruses, a better understanding of their transmission cycles, viewed as a complex multi pathogen-vector-host system is urgently required. Here, we use a spatial datamining framework, based on co-occurrence data that includes biotic niche variables to create models for Dengue, Yellow fever, West Nile Virus and St. Louis encephalitis in Mexico that predict: (i) which mosquito species are likely to be the most important vectors for a given pathogen; (ii) which species are most likely to be important from a multi-pathogenic viewpoint; and (iii) which mosquito and/or mammal assemblages are most likely to play an important role in the transmission cycles. Our predictions are consistent with known information about the dynamics of these mosquito-borne flaviviruses and predict new potential vectors. Our approach can improve disease surveillance efforts and generate useful information regarding public health and biodiversity conservation.

**Abstract:**

Given the significant impact of mosquito-borne flaviviruses (MBFVs) on both human and animal health, predicting their dynamics and understanding their transmission cycle is of the utmost importance. Usually, predictions about the distribution of priority pathogens, such as Dengue, Yellow fever, West Nile Virus and St. Louis encephalitis, relate abiotic elements to simple biotic components, such as a single causal agent. Furthermore, focusing on single pathogens neglects the possibility of interactions and the existence of common elements in the transmission cycles of multiple pathogens. A necessary, but not sufficient, condition that a mosquito be a vector of a MBFV is that it co-occurs with hosts of the pathogen. We therefore use a recently developed modeling framework, based on co-occurrence data, to infer potential biotic interactions between those mosquito and mammal species which have previously been identified as vectors or confirmed positives of at least one of the considered MBFVs. We thus create models for predicting the relative importance of mosquito species as potential vectors for each pathogen, and also for all pathogens together, using the known vectors to validate the models. We infer that various mosquito species are likely to be significant vectors, even though they have not currently been identified as such, and are likely to harbor multiple pathogens, again validating the predictions with known results. Besides the above “niche-based” viewpoint we also consider an assemblage-based analysis, wherein we use a community-identification algorithm to identify those mosquito and/or mammal species that form assemblages by dint of their significant degree of co-occurrence. The most cohesive assemblage includes important primary vectors, such as *A. aegypti*, *A. albopictus*, *C. quinquefasciatus*, *C. pipiens* and mammals with abundant populations that are well-adapted to human environments, such as the white-tailed deer (*Odocoileus virginianus*), peccary (*Tayassu pecari*), opossum (*Didelphis marsupialis*) and bats (*Artibeus lituratus* and *Sturnira lilium*). Our results suggest that this assemblage has an important role in the transmission dynamics of this viral group viewed as a complex multi-pathogen-vector-host system. By including biotic risk factors our approach also modifies the geographical risk profiles of the spatial distribution of MBFVs in Mexico relative to a consideration of only abiotic niche variables.

## 1. Introduction

Predicting the emergence of zoonotic pathogens is a very significant challenge [1]. Usually, predictions about the distribution of priority pathogens relate abiotic elements, such as temperature and precipitation, to simple biotic components, such as a single causal agent (a virus or protozoa) and/or a unique host population (e.g., humans or the Yellow fever mosquito *Aedes aegypti*). For example, most epidemiological studies of the Dengue (DENV) and Zika (ZIKV) viruses have focused only on incidence data in humans, the distribution of the Yellow fever mosquito *A. aegypti* and relationships with abiotic factors [2,3]. Additionally, for mosquito-borne flaviviruses (MBFVs, genus Flavivirus; family Flaviviridae), the role of wildlife species has traditionally been neglected in epidemiological studies. This bias in research and monitoring efforts limits our knowledge about arboviral transmission dynamics, and, therefore, the possibility of predicting the potential distribution of pathogens and the next epidemic and epizootic events.

MBFVs are highly mutable RNA viruses that can readily adapt to new hosts (invertebrates or vertebrates), whose biological characteristics can maintain or alter, in turn, the dynamics of endemic and epizootic transmission cycles [4]. They are a particularly relevant group, as many of them are of importance for human and animal health, impacting welfare, the economy, biodiversity, and ecosystem function and services. For example, the four serotypes of DENV, ZIKV, Yellow fever virus (YFV) and West Nile virus (WNV) affect millions of people [5], while WNV and YFV are known to affect wildlife populations [6,7]. 

In Mexico, and globally, most surveillance and research efforts on MBFVs have traditionally been focused on single-cause cycles. For instance, for DENV (and, more recently, ZIKV), research has mainly focused on *A. aegypti*, or *A. albopictus*, as vectors, and on humans as primary host [8,9,10]. However, due to the impact of global environmental changes, such as human and animal mobilization, habitat fragmentation, climate change, and wild animal trade [11], we might expect to see more outbreaks of MBFVs, such as WNV and ZIKV, as well as a wider set of ecological risk factors. For instance, although humans may be the primary host for several MBFVs, it is worth remembering that a diverse set of MBFVs have been reported in a diverse set of mammals in Mexico, for instance, DENV, St. Louis encephalitis virus (SLEV), WNV and YFV [12,13,14,15,16,17]. The corresponding information for potential avian hosts in Mexico is much sparser, with some number of positives for WNV [18,19,20,21] and very few for SLEV [22,23] having been analyzed. In addition, although there exist other flavivirus of importance, such as ZIKV and Chikungunya (CHIKV), little is known about their sylvatic hosts. For these reasons, given that our analysis will be based on using information of those potential host species that have tested positive for a given MBFV, we will restrict attention in this paper to mammalian species and to DENV, SLEV, WNV and YFV. 

Although, the conclusive detection of MBFVs in wildlife species, using molecular sequencing and viral isolation, is scarce in Mexico and, indeed, elsewhere, there have been reports of DENV in bats, rodents and marsupials from French Guiana and Mexico [12,13,17,24]. Similarly, molecular sequences of YFV and ZIKV have also been identified in primates from South America [25,26], while SLEV isolates were detected in bats (*Tadarida brasiliensis*) from Texas and in armadillos (*Dasypus novemcinctus*) from Brazil [27,28]. Unfortunately, reliable data about the role of mammal hosts in the transmission of MBFVs are difficult to obtain empirically and/or experimentally [29,30]. Several studies of DENV have used bats as study models, but experimentally infected individuals did not replicate or produce antibodies against DENV [31,32]. Elsewhere, other less abundant species, such as armadillos, have been experimentally proposed as potential competent hosts of ZIKV [29].

Given that neither the current distributions of mosquito species, or vertebrate hosts, nor their interactions, are well known, novel theoretical approaches are needed in order to assess the potential role of wildlife in the maintenance and/or spread of MBFVs and to improve our understanding of the dynamics of mosquito-borne diseases. Complex inference networks have emerged as important tool for the study of zoonosis, allowing us to build predictive models of potential vector-host interactions, as well as geographic risk models of pathogen distribution based on species co-distributions [3,33]. This analytical tool has allowed us to redefine the geographic risk profile and host range characterization for vector-borne diseases, such as Leishmaniasis and ZIKV disease [33,34].

## 2. Materials and Methods

### 2.1. Species Data

We searched for information on those mosquito species positive to DENV, YFV, SLEV and WNV in open access data sets [35,36,37,38,39,40]. We also searched reports of mammals that are positive to the same MBFVs in the ISI Web of Knowledge, using “Dengue virus,” “Yellow fever virus,” “St. Louis encephalitis virus,” “West Nile virus,” and “mammal” as keywords [38]. We then built a geographic dataset of point collection data for the 60 mosquito species and 34 wild mammal species that have been confirmed as positive for an MBFV (by serological, molecular or viral isolation tests) and that occur in Mexico using the National Biodiversity Information System (SNIB for its initials in Spanish) [39]. The final dataset contained 40,585 and 92,688 collection points of mosquitos and mammals respectively. 

### 2.2. Data Analyses 

We adopted a nonparametric spatial data mining framework, which allows us to infer potential biotic interactions based on the degree of geographic co-occurrence between species (in our case, mosquito-mammal) [33]. Firstly, we determined the occurrence of species in our study area (continental region of Mexico). In order to obtain a spatial representation of the species distributions, we partitioned our study area with a regular grid that divided the space into regular spatial cells, *X_α_*, of area 15 km by 15 km, and then counted co-occurrences of different taxonomic groups (e.g., mosquito and mammal) within each cell *X_α_* considering *B_i_*(*X*_∝_) = present/not present as a measure of the presence of the mosquito/mammal *i* in the cell *X*_∝_. Our primary objective was to calculate *P*(*B_i_*(*X*_∝_)|***I***(*X*_∝_)), that is, the probability that the distribution measure *B_i_*(*X*_∝_) takes a particular value in the spatial cell *X*_∝_, conditioned on ***I***(*X*_∝_), where ***I***(*X*_∝_) represents the presence or no presence of one or more other species ***I***. For example, ***I*** could represent the presence of mammals reported as positive to the chosen MBFVs. To quantify the relationships between mosquitoes and mammals, we used the probability *P*(*B_i_*|***I***′) = N_Bi∩**I**__′_/*N**_I_***_′_, where N_Bi∩**I**__′_ is the number of spatial cells with co-occurrences of mosquito species *B_i_* and mammal species ***I***′, and *N**_I_***_′_ is the number of cells where the chosen mammals occurred. To evaluate the degree to which co-occurrences are non-random we use the following exact binomial statistical test:(1)εBi|Ik=NIkPBi|Ik−PBiNIk*PBi*1−PBi12
which measures the statistical dependence of *B_i_* on *I_k_*, now representing a particular species, $I_{k}\in I$, relative to the null hypothesis that the distribution of *B_i_* is independent of *I_k_* and randomly distributed over the grid. That is:(2)$$P\left(B_{i}\right)=N_{B_{i}}/N$$
where *N_Bi_* is the number of grid cells with point collections of species *B_i_*, and *N* is the total number of cells in the grid. The sampling distribution of the null hypothesis is a binomial distribution, where, in this case, every cell has a probability *P*(*B_i_*) of containing a collection point of *B_i_*. The numerator of Equation (1) is the difference between the actual number of co-occurrences of *B_i_* and *I_k_*, relative to the expected number if the distribution of collection points was obtained from a binomial distribution with sampling probability *P*(*B_i_*). As the underlying null hypothesis is that of a binomial distribution, it is natural to measure the numerator in standard deviations of this distribution, and this forms the denominator of Equation (1). We interpret the quantitative values of *ε*(*B_i_*|*I_k_*) in the conventional sense of hypothesis testing, by considering the associated *P*-value as the probability that |*ε*(*B_i_*|*I_k_*)| is at least as large as the observed one, and then comparing this *P*-value with a required significance level. In the case where *N_Ik_* is sufficiently large, using the normal distribution as a reasonable approximation to the binomial distribution should be adequate, in which case *ε*(*B_i_*|*I_k_*) = 1.96 would represent the standard 95% confidence interval.

### 2.3. Community Detection and Network Analysis

To build complex inference networks that reflect the inferred associations between mosquitoes, viruses and mammals, we constructed two weighted adjacency matrices with links weighted by *ε* values > 1.96. The first array has 60 rows, representing those mosquito species that are known vectors of a MBFV, and four columns that represent the considered viruses. Individual values of this matrix represent the weighted mean of those ε values associated to the mosquito species in the row i and those mammals reported as positive to the considered MBFVs (see Table 1). For example, to estimate the relationship between *A. aegypti* and YFV, we extract the average of the *ε* values of *A. aegypti* and those mammals reported as positive to YFV (*Tayassu pecari* and *Eira Barbara*, [40]). In this sense, the weighted mean (WE) is calculated as shown in Equation (3), where $Eps_{virus}$ represents the mean *ε* value expressed in columns 2–5 of Table 1 and $\#Mam_{virus}\;$ represents the number of mammal species which have been reported as positive to any MBFV ($\#Mam_{DENV}=14$, $\#Mam_{YFV}=2$, $\#Mam_{SLEV}=13$ and $\#Mam_{WNV}=14$). In the case of *Ae. aegypti*, the WE is calculated as:$$WE=\left(Eps_{DENV}*14+Eps_{YFV}*2+Eps_{SLEV}*13+Eps_{WNV}*14\right)/43,\;$$
where $Eps_{DENV}$, $Eps_{YFV}$, $Eps_{SLEV}$ and $Eps_{WNV}$ are the values of *ε* between *A. aegypti* and each pathogen as proxied by its mammal hosts. The second matrix was constructed using the *ε* values of each mosquito (row) and mammal (column) pair. So, values $\left(i,k\right)$ of this matrix represent directly the $\varepsilon \left(B_{i}|I_{k}\right)$ values. We then selected only those values of $\varepsilon \left(B_{i}|I_{k}\right)$ > 1.96 corresponding to the standard 95% confidence interval.
(3)$$WE=\frac{\sum_{i=virus}Eps_{i}*\left(\#Mam_{i}\right)}{\sum_{i=virus}\#Mam_{i}}\;$$

We then performed a community detection analysis of both networks, using the Louvain algorithm for the detection of communities in large networks [41]. The method consists of recurrently merging communities that optimize the degree of modularity as an objective function with which to optimize the network. In the case of weighted networks, it is defined in [41] as:(4)$$Q=\frac{1}{2m}\underset{i,k}{\sum }\left[A_{i,k}-\frac{V_{i}H_{j}}{2m}\right]\delta \left(c_{i},c_{k}\right)\;$$
where $A_{i,k}$ represents the weight of the edge (*ε* value) between mosquito *i* and mammal *k*; $V_{i}=\sum_{k}A_{i,k}$ is the sum of the weights of the edges attached to the mosquito species *i*; $H_{k}=\sum_{i}A_{i,k}$ is the sum of the weights of edges attached to the mammal species *k*; $c_{i}$ is the community to which mosquito *i* is assigned; the δ function $\delta \left(\upsilon ,\nu \right)$ is 1 if υ=ν and 0 otherwise; and $m=\frac{1}{2}\sum_{i,k}A_{i,k}$. Thus, the Louvain algorithm identifies those mosquito communities that better represent the modularity of the underlying ecological network.

A hierarchical analysis of the species assemblages was performed by using structural cohesive and embeddedness methods [42]. Structural cohesion is defined as the minimum number of nodes which, if removed from a group, would disconnect the group. In this regard, a collection of nodes is structurally cohesive to the extent that the relations of its members hold it together. The identification of communities by their structural cohesion is a process of cohesive blocking. Identification of the cohesive blocks involves a recursive process: One first identifies the k-connectivity and the path connectivity of an input graph, then removes the k-cutset(s) that hold(s) the network together. One then repeats this procedure on the resulting subgraphs, until no further cutting can be completed. As such, any $\left(k+l\right)$-connected set embedded within the network will be identified. Moreover, each procedure’s iteration takes us deeper into the network, as weakly connected nodes are removed first, leaving stronger connected nodes and stronger connected sets. As a consequence, it uncovers the nested structure of cohesion in a network. Lastly, the implementation of this analysis was performed using the algorithm described by [42]. 

### 2.4. Predictive Models and Risk Maps

There are different ways in which predictive models may be generated from co-occurrence data. Firstly, we generated predictive models to infer which mosquito species are most likely to be vectors of a given MBFV, as well as those species most likely to be vectors of multiple MBFVs. The intuition here is that those mosquito species with the most statistically significant degree of co-occurrence with those mammal species which have been confirmed positive for the MBFV are most likely to be vectors.

Although co-occurrence between a mosquito and infected mammals is a necessary condition that the mosquito become infected, it is not sufficient. For instance, the mammal may not be a competent host, or the mosquito may not preferentially feed on that mammal. However, such co-occurrence models serve as a base model which can then be compared to known results. We may thus determine if the known vector species are associated with particularly high values of *ε*. A suitable null hypothesis for testing the predictive performance of such a model is that known mosquito species are randomly distributed in the list of mosquito species ranked by *ε*. Thus, if a regression of the number of positive mosquito species against *ε* yields a significant regression coefficient then we determine that this is not consistent with the null hypothesis and therefore *ε* is a predictor of the likelihood that a given mosquito species is a vector of multiple pathogens or not. Explicitly, we divide the ranked list of mosquito species into deciles and regress the percentage of known vector species in each group against the average value of ε for those species in the decile.

Similarly, a simple predictive model for those mosquito species most likely to be associated with multiple MBFVs considers the weighted sum of *ε* values, as seen in Equation (3), as this is a measure of the degree of co-occurrence between a mosquito species and all mammal species identified as confirmed positives of any of the MBFVs. In this case, as with an individual MBFV, we may divide the list of mosquito species into deciles and then regress, for example, the number of known MBFVs associated with a given mosquito species as a function of the average value of the weighted mean of ε values for the mosquitos in that decile. A statistically significant regression coefficient then implies that the weighted mean of *ε* values is a predictor of the number of MBFVs associated with a given mosquito species.

It is worth pointing out here that another simple predictive model could be based on the notion that those mosquitos most likely to be vectors of a given pathogen, or of multiple pathogens, are those with the widest geographic distribution. The intuition here is that if hosts were randomly distributed with respect to the mosquito distributions, then one would expect to have a higher probability of infection for those mosquitos with the widest distribution. If this is not the case, it is circumstantial evidence that the associations between those mammals confirmed as positive and their potential vectors is non-random.

A next step is to identify those geographic regions with suitable ecological conditions for MBFV presence. A risk map for a particular pathogen should consider the presence of each biotic agent that is potentially involved in its transmission cycle, i.e., each vector-confirmed positive interaction. In order to determine those spatial regions with a higher risk for exhibiting mosquito-mammal interactions, we modelled the potential distribution of each mosquito species based on the presence data for each co-occurring mammal using the Score function, $S\left(B_{i}|I\right)$, proposed in [43]; where $S\left(B_{i}|I\right)$ is a measure of the probability to find a presence of the variable $B_{i}$ (i.e., a mosquito species) when the mammal profile is ***I***. As this score function can be calculated for each spatial cell, $X_{\propto }$, this approach allows us to find those cells where vector and potential mammal hosts are most likely to co-occur and, therefore, potentially interact.

To build a predictive model for a particular MBFV, each individual mosquito species distribution model was used as a classifier, where score values > 0 are classified as corresponding to potential mosquito presence, and on the contrary if the score is < 0. Finally, individual maps were combined considering mosquito assemblages identified for each MBFV in the network analysis.

## 3. Results

We identified in the literature 60 mosquito and 34 wild mammal species reported as positive to the considered MBFVs (Appendix A). We ranked those mosquito species that could potentially harbor known MBFVs in Mexico (Table 1) based on their co-occurrence with mammals that have been confirmed as positive for each MBFV as proxied using *ε*. As mentioned, the intuition here is that—all else being equal—those mosquitoes with the highest degree of co-occurrence with infected mammals will be those most likely to be infected by taking blood meals from those mammals and therefore the most likely to be infected when tested. We emphasize here the “all else being equal” caveat. There are of course, other reasons why mosquitoes and mammals may co-occur and so the identification of a mosquito-mammal interaction may be confounded. Among these are shared fundamental niches, though, physiologically, a mammal and a mosquito are so different, the confounding is not likely to be great. A more relevant confounder would be the presence of avian hosts, whereby the co-occurrence relation between mammal and mosquito was indirect or represented a spillover. We have stated why in the present study we have not included birds. However, this subject deserves to be studied in the near future. 

In this way, each column in Table 1, when ordered from highest to lowest ε values, is a prediction model for those mosquito species that are most likely to be vectors of the considered pathogens. Similarly, using the grouped adjacency matrix, by calculating the weighted mean of *ε* values for each mosquito species, we may order by this weighted mean and use this ordered list as a prediction model for those mosquitos most likely to be vectors of multiple MBFVs.

In terms of the weighted mean of *ε*, at the top of the ranked list, we observed those mosquito species that are widely distributed throughout Mexico, and which therefore, by dint of their sample geographical distribution, allow for a significant potential interaction with the confirmed mammal hosts of the different MBFVs. Conversely, at the bottom of the list, we find those mosquito species with reduced geographical distributions that therefore limit potential spatial interactions with the known mammals confirmed as positive. Moreover, at the top of the ranked list, we also observed that the majority of mosquitoes were confirmed as being positive to two or more pathogens (see Figure 1). However, the fact that the most widely distributed mosquitos are highly ranked does not mean that a model based on ranking according to distributional area is as predictive as our model based on ranking according to *ε*. To illustrate this, we estimate the proportion of the sample distribution area for each mosquito (Table 1, column 7), and then determine the Pearson correlation between the two model types for each MBFV (Table 1, columns 3–7). The correlation coefficients for the two models for each MBFV are: $\rho_{DENV}=0.485$, $\rho_{YFV}=0.078$, $\rho_{SLEV}=0.692$, $\rho_{WNV}=0.760$ and $\rho_{WeightdMean}=0.651$. This indicates that the models are quite distinct and that the pattern of positive mosquito species for each MBFV is not determined by the pure random co-occurrence between mosquitos and mammals, as proxied by their relative distributional areas. This is particularly the case for DENV and YFV where the correlation coefficients are lower than the others.

Testing each prediction model using a regression analysis, we assessed the hypothesis that confirmed vectors are in the top deciles of our ranked lists for each MBFV, and therefore, that those mosquitoes in the top deciles are the most likely potential candidates to harbor the corresponding virus. Table 2 shows the regression coefficient and corresponding t and p-values for each model. For DENV, YFV and SLEV we note that the proportion of confirmed vector species significantly increases towards the higher deciles and, thus, is a predictor of which mosquito species are most likely to be vectors. The number of previously confirmed vectors for these MBFVs were 2 (DENV), 3 (YFV) and 13 (SLEV), respectively. However, our results indicate that for these MBFVs this number is most probably an underestimate. Conversely, for WNV there is no significant relation between the proportion of confirmed vectors and higher ε values. However, this result is to be expected, given that 53 of the 60 mosquitoes assessed here are positive to WNV and, consequently, the proportion of confirmed mosquito species must be similar throughout the deciles of the ranked list.

Finally, we tested if the weighted sum of *ε* values and the size of distribution area are able to predict which species are the most likely to be associated with multiple MBFVs. For these multi-pathogen models, we expect that those mosquitos associated with more than one virus should be in the top deciles of the lists ranked by weighted *ε* or by the size of the mosquitos’ ranges. In distinction to previous studies, where statistical differences based on ε values and species size ranges have been noted in the case of single-pathogen models [43], here we found in both cases a significant linear relationship (Table 2), therefore showing that both higher *ε* values and range sizes are significant predictors of those vector species which harbor multiple MBFVs. This result highlights the fact that, in a multi-pathogen model, those vector species with wide geographic distributions are more likely to be in contact with different pathogens.

Community analysis identified four mosquito communities based on their spatial proximity to those mammals confirmed as positive to each MBFV. The community associated with DENV and SLEV (formerly named DENV/SLEV) is composed of 16 mosquito species. Nine of these are significantly linked with mosquito communities that are associated with another three MBFVs (Figure 2, red nodes). In this regard, *Aedes aegypti*, *Culex quinquefasciatus*, *C nigripalpus*, *C coronator*, *C erraticus*, *Psorophora howardii* and *Haemagogus mesodentatus* are also linked with the YFV and WNV communities, *A taeniorhynchus*, *Psorophora ferox*, *Mansonia titillans* and *Uranotaenia lowii* maintain links to the YFV community and *A trivitatus* and *A albopictus* are related to WNV. The YFV mosquito community contains 13 mosquito species (Figure 2, blue nodes). *Anopheles crucians* and *Aedes scapularis* are also connected with both DENV/SLEV and WNV communities. Nine of these mosquitos are also members of the DENV/SLEV community (*Deinocerites cancer*, *C taeniopus*, *C bahamensis*, *C habilitator*, *A infirmatus*, *A atropos*, *Sabethes chloropterus*, *A fulvus* and *A atlanticus*). The WNV community contains 14 mosquito species (Figure 2, green nodes). Three *Culex* spp. are linked to both DENV/SLEV and YFV communities (*C pipiens*, *C quinquefasciatus* and *C salinarius*), seven species are related to the DENV/SLEV community (*A punctipennis*, *C thriambus*, *C tarsalis*, *C stigmatosoma*, *C restuans* and *C particeps*), and only *A triseriatus* is related to the YFV community. Lastly, the “not linked” mosquito species form a group that does not co-occur in any significant way with those hosts confirmed as positive for any of the considered MBFVs. These nodes are not presented in Figure 2.

Next, the cohesion analysis applied to each subnetwork (as described above) allows us to identify the most connected community for each MBFV. In this regard, the DENV, SLEV and WNV subnetworks have the same species community, with the most significant degree of cohesiveness consisting of 41 members (20 and 21 mosquito and mammal species, respectively; see Figure 3a,c,d, red nodes). The YFV subnetwork has a cohesive community (Figure 3b) with the same number of members (41 nodes represented by 20 and 21 mosquito and mammal species, respectively). The difference between the cohesive blocks of the two communities are just two nodes: *A Condolescens*, which appears in the DENV, SLEV and WNV cohesive communities, but not in the YFV community, and *D Cancer*, which belongs to the YFV community, but not the others. Finally, using the common nodes of the four cohesive communities (most cohesive block), we identified the number of MBFVs confirmed for each mosquito and mammal species (Figure 4). There, we observed that 37.5% of the 40 species are positive for two or more MBFVs.

Regarding our distribution maps, we considered the spatial distribution of the 34 confirmed mammals. Then, we observed that the first four mosquito species at the top of our ranked list have quite similar spatial distributions (see Figure 5). As mentioned above, these mosquito species are spatially connected with several MBFV nodes (see Figure 2). We then overlapped the prediction maps of each mosquito species for the three mosquito communities associated with the MBFVs´ nodes, described in Figure 2 (DENV/SLEV, WNV and YFV; Figure 6a–c). In this way, we observed that the spatial distribution of mosquito species richness is similar for the three communities associated with the MBFVs. Using a standardized scale, we notice that the higher risk zones are located at both Mexican coasts (Atlantic and Pacific) and in Central Mexico. Finally, we observed that the spatial distribution of risk for the different MBFVs, associated with their corresponding mosquito communities, is similar. The validation of these predicted distributions requires data associated with the spatial distribution of cases of these MBFVs in Mexico. Unfortunately, for YFV there have been no published cases and limited data are available for SLEV and WNV. 

## 4. Discussion

Active surveillance and current research on mosquito vectors of MBFVs is mostly focused on the most abundant species in urban settings, *A aegypti* and *C quinquefiasciatus*, and on humans as hosts [8,44,45]. However, we should not underestimate the importance in viral transmission of other vector and host species, as well as the communities that they form [46]. It is essential to improve mosquito management strategies, as well as predict, prevent and control diseases caused by MBFVs. An important task is to predict which mosquito species are potentially the most important vectors, both for a particular pathogen as well as for multiple pathogens. In this case, Table 1 describes six different prediction models, four single-pathogen models (Columns 3–6), one multi-pathogen model (Column 7) and one model based only on mosquito distribution area (Column 8). 

The predictive single-pathogen models classify mosquitoes based on the estimation of the potential risk area of each pathogen given the known information about MBFV-confirmed positive mammals. In this sense, this classification is associated with the number of confirmed MBFV-confirmed positive mammals and their distribution area. Thus, one might expect that those mosquitoes with a larger distributional area would be those most likely to be infected. However, we clearly see that this is not the case, and is particularly so for YFV and DENV. For example, for the YFV model, using only the two known confirmed positive mammals, the ranked list by *ε* is completely different to the list ranked by mosquito species distributional area. We therefore infer that the non-random co-occurrence between vector and host is due to some underlying interaction, such as the fact that the mosquito feeds on these mammal species, or that there is some underlying abiotic factor that links the vector and these mammal species. The correlation between the ε-based and distributional area-based models for DENV also exhibits relatively little correlation (0.485), in spite of the fact that there are 14 identified hosts, the same as WNV where, in the latter, the correlation coefficient between the two models is 0.760. We interpret this difference as evidence that DENV is more specialized in its transmission cycle to particular mosquito-mammal pairs, whereas WNV is more permissive, being preferentially linked to those mosquitoes that have the widest distributional area.

On the other hand, we see that the multi-pathogen model using the mean value of *ε* leads to very similar results to that of the distributional model, with a correlation coefficient of 0.651 between the two lists. We interpret this as being due to the fact that the confirmed positive mammals taken across all four MBFVs represent a very diverse and heterogeneous set of niche conditions. In this circumstance, one would not expect to see a specialized ecological relationship between a given mosquito species and these many mammal species, as the latter do not possess any relevant characteristic that could represent a basis for such a specialization. In this case then one would expect mosquito distributional area to be a good predictor of probability of being a vector for multiple pathogens. 

Interestingly, our ranking is consistent with known evidence of these mosquito species as potential vectors of different pathogens. The most studied species for example, *A aegypti*, was the first ranked species in our list, and is well known for its capacity to transmit all our considered MBFVs, as well as other pathogenic viruses [47,48]. The second-ranked mosquito species, *C coronator*, inhabits rural areas [49]. However, its ecology and epidemiology are poorly known. With our approach, we note that it is strongly associated with mammals confirmed as positive for DENV, YFV and SLEV, which leads us to hypothesize that it is a likely vector of one or more of these MBFVs. As rural and synanthropic species can invade natural areas, taking advantage of human behaviors that promote favorable sites for reproduction (e.g., garbage management), its monitoring should be extended to urban, rural and natural areas. The third in our ranked list, *C quinquefasciatus* (Table 1 and Figure 1), is the most abundant species in Mexican urban settings [50]. It feeds on both birds and mammals (principally on humans and dogs [50]), and therefore may transmit different pathogens (such as WNV and ZIKV [51]) among a wide diversity of hosts. The fourth in our ranked list is *C nigripalpus*, from which SLEV was isolated in Southern Mexico [22]. The fifth in our ranked list, *H mesodentatus*, is highly ranked as a vector of DENV, YFV and SLEV (Table 1), but much less so for WNV. In fact, YFV was isolated from this mosquito species in Guatemala [52], and currently, DENV serotype 1 was detected in its conspecific *H leucocelaenus* in Brazil [53]. Therefore, our approach accurately suggests the potential role of different mosquito species in a multi-pathogenic framework.

Figure 2 shows the mosquito assemblages linked to each MBFV. This network shows that 37 mosquitoes out of 60 (61.67%) are potentially associated with two or more MBFVs. In spite of the limited virological evidence in this case, 6 out of 60 mosquitoes (16.67%) have been confirmed as positive to two or more pathogens. It is interesting to note that the community with the highest mosquito richness (with 16 species) merge both DENV and SLEV as both viruses share five hosts (J = 0.2273, Jaccard index). This mosquito community includes the most abundant and studied species in Mexico, such as *A aegypti*, *A albopictus*, *C nigripalpus*, *C coronator* and *P ferox* [22,37,49,54]. This mosquito community also includes *C erraticus*, which is one of the most abundant species in temperate regions [55]. It also includes *H mesodentatus* and *Psorophora howardii*, that inhabit sylvatic sites which can be linked to potential wild mammal hosts [54,56]. With our approach, we see that the DENV/SLEV community involves nine mosquito genera, including both anthropophilic and non-anthropophilic mosquitoes (e.g., *A aegypti* and *Uranotaenia* sp., respectively) and which inhabit distinct habitats, such as urban, sub-urban and pristine areas (Figure 2). So, although DENV is well established in urban settlements, where it circulates mainly between humans, *A aegypti* and *A albopictus* [37]; in other tropical countries, other *Aedes* spp. have been linked to DENV sylvatic cycles [57], while other genera (*Sabethes* and *Culex*) have been incriminated by molecular findings [58,59]. These results are consistent with our prediction that the DENV transmission cycle is much more complex and diverse than that suggested by its link to *A aegypti* and *A albopictus* in urban environments. Similarly, SLEV, which originated from Central America and circulates mainly between *Culex* mosquitoes and avian or mammal hosts [22,60], is predicted to have a complex transmission cycle, with multiple vectors and hosts involved. Together, our findings suggest that DENV and SLEV are more generalist than had been previously thought.

Turning now to a discussion of the complex inference networks for MBFVs, we noted that the mosquito communities linked with WNV and YFV have lower species richness than the DENV/SLEV community, with just 14 and 13 species, respectively (Figure 2). The WNV community includes two of the primary vectors of this MBFV, *C quinquefasciatus* and *C pipiens*. It also includes ornithophilic species (*C quinquefasciatus*, *C pipiens* and *A punctipennis*) that alternate their blood meals between avian and mammalian hosts [61]. It is well known that WNV infects a wide range of mosquito genera, and other hosts, including sandflies, ticks, reptiles and amphibians [36,62,63,64] and has been reported in most parts of Mexico, including both urban and natural areas. So, there are many mosquito and avian species, as well as domestic animals, incriminated in its transmission cycle [18,45,65], though current knowledge about the species involved is insufficient and there are many gaps in our knowledge of WNV dynamics.

The YFV mosquito assemblage includes *S chloropterus*, which has been incriminated in YFV circulation in South America [35]. Besides *Aedes* mosquitoes, this community also includes other genera that may be involved in YFV transmission, but have not yet been tested (e.g., *Deinocerites*). Circulation of YFV in Mexico has not been confirmed in mosquitoes, and, regarding hosts, there are only reports of antibodies against this virus in bats and urban rodents from Merida city, in the Yucatan Peninsula [15,16]. However, these results may represent cross-reactions with other circulating MBFVs within serological tests [15,16]. YFV in humans was eradicated from Mexico many decades ago, but it still circulates in Central America and various Caribbean islands within primates and arboreal mosquitoes [66]. Our results suggest that we should consider a potentially wide range of vectors and hosts when evaluating the risk of reintroduction of YFV into Mexico. Currently, the study of arboreal mosquito communities is neglected, and entomologic studies in pristine areas, where many potential hosts can be found, are scarce [54].

The “disconnected” assemblage (mosquitoes not shown in Figure 2) includes mosquito species that are at the bottom of our ranked list and therefore with relatively lower levels of co-occurrence with infected mammals. However, their potential role in MBFVs´ transmission cycles should not be completely discarded. For example, *A vexans* has been found to be positive for ZIKV, and therefore it is essential to consider it as a potential multi-pathogen vector [51]. In fact, we should not a priori underestimate the direct or indirect participation of any species (vector or host) in the MBFVs´ dynamic but, rather, use approaches such as ours to prioritize disease surveillance efforts, test new hypotheses and develop actionable information regarding public health and biodiversity conservation. Furthermore, recognizing how mosquito communities modify MBFVs’ transmission cycles is important to be able to infer and predict viral dynamics [67]. 

In Figure 3, we observed that the direct connectivity of the subnetwork associated to DENV includes 82.98% (78 out of 94) of all nodes (see STATUS on Figure 3a). Meanwhile, the YFV subnetwork contains 60.63% of all species (see STATUS Figure 3b), of which only five species (three and two mosquito and mammal species, respectively) were confirmed as positive to YFV. Similarly, the SLEV and WNV subnetworks include 96.81% and 98.94% of all nodes, respectively (see STATUS Figure 3c,d). The WNV subnetwork shows that 69 species (16 and 53 mammal and mosquito species, respectively) are confirmed as positive to WNV. Here, we constructed a cohesive community, which describes mosquito-mammal assemblages with a high degree of spatial correlation. We observed that all subnetworks have similar connectivity paths and contain the most highly connected nodes (as describe in Results section). The common members of the most cohesive assemblage of all pathogens (Figure 4) includes the primary mosquito vector *A aegypti* as well as other well-known species, such as *C quinquefasciatus*, *C pipiens* and *A albopictus*, among others. It also contains mammals with abundant populations, that are well-adapted to human environments, such as the white-tailed deer (*Odocoileus virginianus*), peccary (*Tayassu pecari*), opossum (*Didelphis marsupialis*) and bats (*Artibeus lituratus* and *Sturnira lilium*). Opossums and bats have also been confirmed as positive for two of our four MBFVs [13,16,17,68]. In this regard, we may consider the most cohesive network as a “core group” of those potential hosts involved in MBFVs in Mexico, and also emphasizes the importance of considering transmission cycles from a multi-MBFV-vector-host perspective.

To our knowledge, there are no risk maps that predict the distribution of mosquito vectors or of multiple MBFVs in Mexico. At a regional level, geographical information has been restricted to distribution maps for a few mosquito species, such as *A cozumelensis*, *C quinquefasciatus* and *C coronator* [49]. At the country level, most of the information is in the form of incidence maps for DENV based on human cases [8]. An exception is found in [28], who used the current approach but analyzed only the potential distribution of the single vector *A aegypti* and its co-occurrence with Mexican wild mammals. In this regard, we extended the approach of [34] to estimate the distribution of pathogens through the richness of their potential vectors, where the latter considers the extraction of communities based on the MBFV’s distributions from those mammals confirmed as positive as a proxy for the distribution of the pathogen itself. For example, Figure 5 shows the potential distribution of *A aegypti*, a well-known predictor for the distribution of DENV, based on the co-occurrence of these 34 known confirmed positive mammals, but this prediction does not consider the potential role of other mosquito species. However, by including the mosquito assemblages of our community network analysis (Figure 2), we can estimate the distribution of MBFVs based on the distributions of those vectors which have been identified as positive to at least one MBFV. So, mosquito richness can be used as a proxy with which to predict the probability to find MBFVs in a particular area. In this sense, Figure 6a shows a prediction of the distribution of DENV and SLEV given by the community of multiple vectors and multiple confirmed positive mammals. Comparing these two distributions (Figure 5 and Figure 6a) we can see that the distribution of *A aegypti* is limited and it does not reveal an appropriate pathogen distribution, but our model shows a more inclusive distribution. Moreover, Figure 6b,c describe the potential distribution of the YFV and WNV communities, respectively. Both predicted distributions are similar to the DENV/SLEV distribution, but with the WNV distribution showing a trend towards northern states of Mexico. Within this complex scenario, the potential distribution of MBFVs is expanded. This is especially the case for poorly known vector and host species, which have generally been neglected by health authorities, but which could play an important role in those enzootic cycles of zoonotic pathogens that are essential for the maintenance of MBFVs in nature and this is relevant information for the monitoring of arboviral diseases and their vectors.

Our analysis did not consider some other confirmed positive mammals as potential hosts due to the fact that they are poorly represented in the data set, including rodents of the genera *Rattus* and *Mus*, in spite of the fact that they are associated with MBFVs in urban and rural sites in Mexico [15,65,69]. Similarly, for domestic mammals and other potential non-mammalian warm-blooded hosts [65]. The presence of domestic animals in natural habitats modifies the transmission of vector-transmitted pathogens, so their potential role in the transmission cycle deserves further study [70]. Our approach considers the co-occurrence among potential vectors and wild mammals that have been confirmed positive, but it does not guarantee that the species interact with each other, as co-occurrence is a necessary but not sufficient condition for interaction. For example, the fact that mosquito females might feed on these mammals does not itself prove MBFV transmission. So, we need to confirm a trophic interaction between mosquitoes and potential wild hosts in conserved and disturbed sites in order to evaluate the potential participants within enzootic and epizootic transmission cycles [71]. Most of the mosquito species listed are anthropophilic, but they can feed on any available warm-blooded hosts [72]. However, small size mammals are not usually a typical food source among mosquitoes [58,73], while other mosquito species seem to not bite humans, such as *Uranotaenia sapphirina* and *Ur. lowii*, that feed on amphibians [74], or *Orthopodomyia alba* that feeds on wild birds [75]. As the choice of a blood meal by mosquitoes depends on the current availability of warm-blooded hosts, in this paper we do not exclude any potential trophic interactions between mosquitoes and mammals.

Although our approach is capable of predicting and understanding the transmission cycles of any MBFV, along with any biotic agents whatsoever, it does require the corresponding data for both the MBFVs and the agents. It is for that reason we restricted attention to DENV, YFV, SLEV and WNV and to only potential mammalian hosts, as there have been multiple studies that have confirmed an ample number of mammal species as being infected with these MBFVs. By this we do not wish to imply that avian hosts are not important, especially in the case of WNV and SLEV. We will return, in the future, to a consideration of both mammalian and avian potential hosts so as to make a potential comparison of their relative importance in the transmission cycle of these MBFVs and, importantly, their potential role from a multi-pathogenic perspective. Additionally, our methodology is capable of disentangling causal chains so as to better understand the role of confounders [76,77]. This is particularly relevant in the case of mammals and birds as their co-occurrence with MBFVs may be intermediated by mutual interactions other than direct interactions with mosquitos.

Our approach in this paper combines both recently developed analytical tools and empirical knowledge about potential vectors and hosts. However, its application must be long-term, active and empirically tested, due to the fact that mosquito and host diversity and range, and their associated distributions, are quite dynamic. Additionally, current global changes, most of which are of anthropogenic origin, are modifying viral dynamics. We must therefore be prepared for future pathogen emergence and invasions. Finally, it is urgent to better understand and quantify the current distribution of MBFVs in Mexico in order to improve the decision-making process in these areas: (a) wild mammal conservation priorities; (b) mosquito management; and (c) prediction, prevention, and control of vector-borne diseases. Finally, our approach emphasizes the importance of designing and using new theoretical models to better understand the dynamics of MBFVs and, in particular, to view and model them from a multi-pathogenic point of view.

## Figures and Tables

**Figure 1 insects-12-00398-f001:**
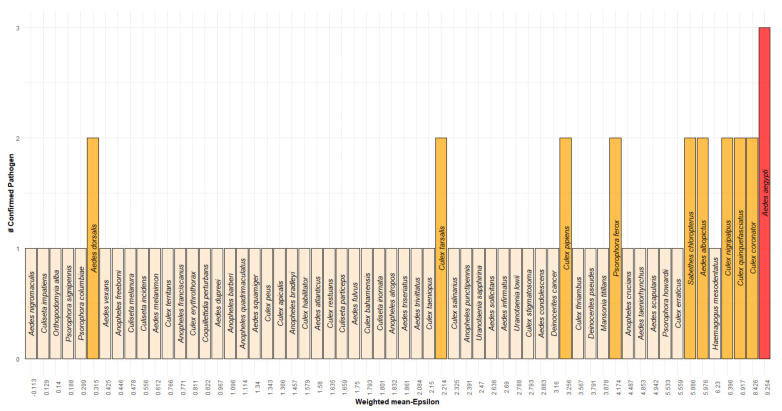
List of vectors ranked by weighted mean ε. X-axis: Weighted mean ε. Y-axis: Number of confirmed pathogens per mosquito.

**Figure 2 insects-12-00398-f002:**
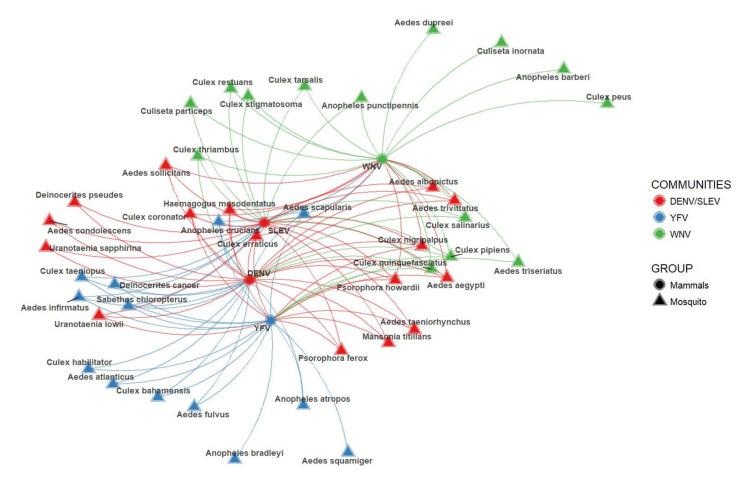
Louvain community detection by grouping mammals per pathogen.

**Figure 3 insects-12-00398-f003:**
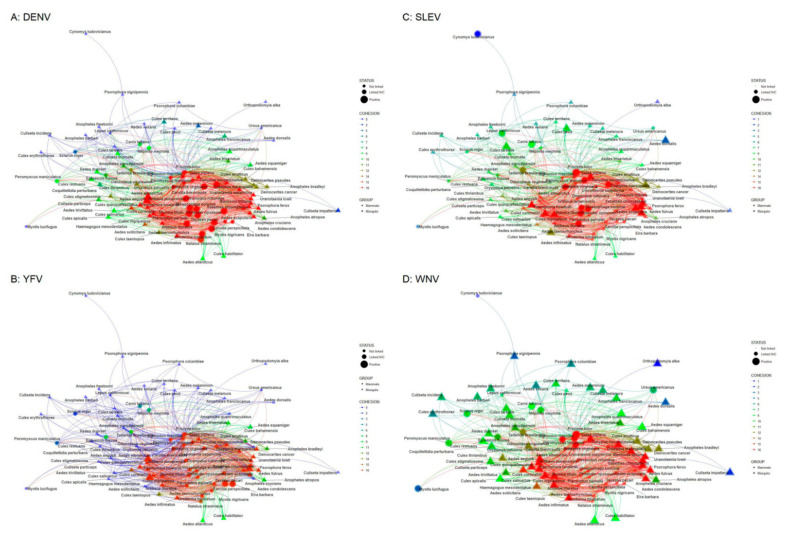
Cohesion analysis by path detection. STATUS: represents the confirmation and path relationship between nodes (Positive: Nodes reported as positive to MBFV, Linked N/C: Nodes linked to a positive node but not confirmed, not linked: Nodes without an Edge to any positive node). COHESION: Cohesion value (greater value implies more cohesion). GROUP: type of node (circle: mammals, triangle: vectors). (**A**): Dengue (DENV), (**B**): Yellow fever virus (YFV), (**C**): St. Louis encephalitis virus (SLEV) and (**D**): West Nile virus (WNV).

**Figure 4 insects-12-00398-f004:**
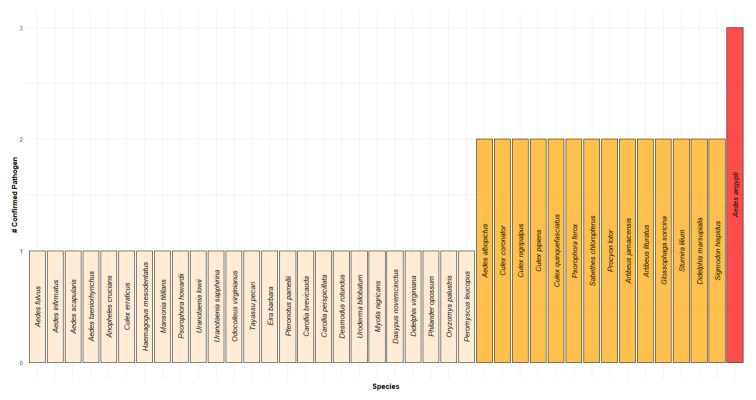
Cohesive block: Common member of the most cohesive blocks of the four pathogens. X-axis: Mosquitoes and mammal species. Y-axis: Number of confirmed pathogens per species.

**Figure 5 insects-12-00398-f005:**
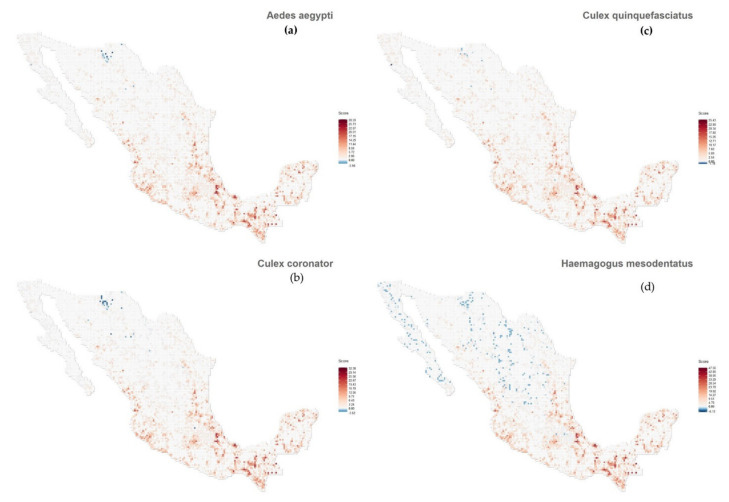
Predicted map distribution. (**a**): Aedes aegypti, (**b**): Culex coronator, (**c**): Culex quinquefasciatus and (**d**): Haemagogus mesodentatus.

**Figure 6 insects-12-00398-f006:**
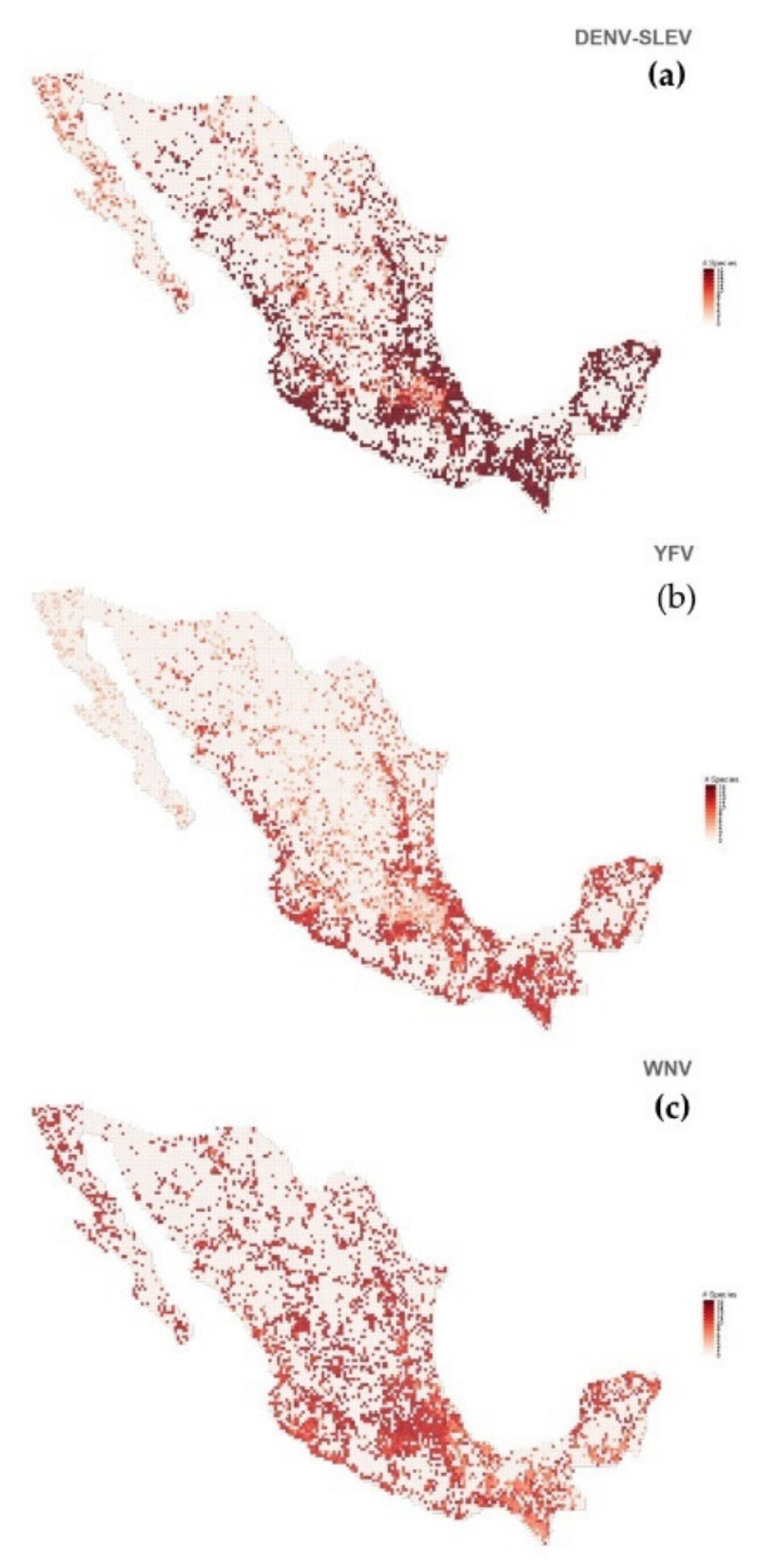
Predicted richness map distributions. (**a**): Dengue virus and St. Louis encephalitis virus community. (**b**): Yellow fever virus community. (**c**): West Nile virus community.

**Table 1 insects-12-00398-t001:** List of known vectors for the four mosquito-borne flaviviruses (MBFVs) considered and their weighted ε values calculated from the co-occurring positive mammal species for each MBFV and for all together.

Ranking	Species	Epsilon DENV	Epsilon YFV	Epsilon SLEV	Epsilon WNV	Weighted Mean	Sample Area Proportion	# of Confirmed Pathogens
1	*Aedes aegypti*	* 11.148	* 2.969	* 10.568	* 7.038	9.254	0.0697	3
2	*Culex coronator*	11.132	5.773	* 9.632	* 4.978	8.426	0.0307	2
3	*Culex quinquefasciatus*	7.217	2.898	* 8.028	* 6.161	6.917	0.0517	2
4	*Culex nigripalpus*	9.670	7.122	* 6.573	* 2.853	6.396	0.0094	2
5	*Haemagogus mesodentatus*	9.864	* 4.910	6.535	2.500	6.230	0.0027	1
6	*Aedes albopictus*	* 7.300	1.296	6.566	* 4.772	5.976	0.014	2
7	*Sabethes chloropterus*	10.121	* 9.354	* 5.361	1.643	5.886	0.0021	2
8	*Culex erraticus*	7.003	6.018	6.258	* 3.401	5.559	0.0067	1
9	*Psorophora howardii*	7.112	2.919	6.204	* 3.704	5.533	0.0012	1
10	*Aedes scapularis*	6.887	8.498	* 5.05	2.389	4.942	0.0053	1
11	*Aedes taeniorhynchus*	7.537	3.808	5.361	* 1.847	4.853	0.0072	1
12	*Anopheles crucians*	6.565	6.230	* 4.653	2.005	4.487	0.0033	1
13	*Psorophora ferox*	6.641	3.917	* 3.947	* 1.953	4.174	0.0039	2
14	*Mansonia titillans*	6.262	3.735	4.037	* 1.368	3.878	0.0028	1
15	*Deinocerites pseudes*	5.277	−0.399	* 5.040	1.746	3.791	0.0021	1
16	*Culex thriambus*	3.019	−0.025	4.546	* 3.719	3.567	0.0065	1
17	*Culex pipiens*	3.025	4.068	* 3.379	* 3.255	3.256	0.0024	2
18	*Deinocerites cancer*	3.887	3.692	3.632	* 1.918	3.160	0.0008	1
19	*Aedes condolescens*	5.169	−0.028	2.954	*0.946	2.883	0.0001	1
20	*Culex stigmatosoma*	1.569	−0.688	3.597	* 3.767	2.793	0.0319	1
21	*Uranotaenia lowii*	3.669	2.637	3.663	* 1.117	2.788	0.0029	1
22	*Aedes infirmatus*	4.305	5.954	2.691	* 0.608	2.690	0.0007	1
23	*Aedes sollicitans*	2.985	1.205	3.100	* 2.061	2.636	0.004	1
24	*Uranotaenia sapphirina*	3.693	0.884	3.045	* 0.941	2.470	0.0019	1
25	*Anopheles punctipennis*	0.649	−0.690	2.656	* 4.327	2.391	0.0059	1
26	*Culex salinarius*	2.239	1.657	2.604	* 2.248	2.325	0.0027	1
27	*Culex tarsalis*	−0.109	−0.243	* 2.681	* 4.454	2.214	0.0086	2
28	*Culex taeniopus*	2.738	3.256	* 2.682	0.910	2.150	0.001	1
29	*Aedes trivittatus*	2.077	−0.426	2.568	* 2.001	2.084	0.0023	1
30	*Aedes triseriatus*	0.793	3.256	1.868	* 2.724	1.861	0.001	1
31	*Anopheles atropos*	2.463	5.839	1.552	* 0.889	1.832	0.0001	1
32	*Culiseta inornata*	0.245	−0.426	1.942	*3.545	1.801	0.0023	1
33	*Culex bahamensis*	2.075	5.693	1.67	* 1.067	1.793	0.0004	1
34	*Aedes fulvus*	2.75	5.056	1.125	* 0.859	1.750	0.0004	1
35	*Culiseta particeps*	0.38	−0.496	1.995	* 2.934	1.659	0.0123	1
36	*Culex restuans*	0.844	0.459	1.995	* 2.261	1.635	0.0033	1
37	*Aedes atlanticus*	3.016	5.732	1.195	* −0.093	1.579	0.0001	1
38	*Culex habilitator*	3.016	5.732	1.195	* −0.093	1.579	0.0001	1
39	*Anopheles bradleyi*	1.281	3.442	1.876	* 0.962	1.457	0.0009	1
40	*Culex apicalis*	1.25	−0.238	1.772	* 1.334	1.366	0.0008	1
41	*Culex peus*	0.283	−0.28	*1.525	2.465	1.343	0.0011	1
42	*Aedes squamiger*	1.16	5.732	1.305	* 0.925	1.340	0.0001	1
43	*Anopheles quadrimaculatus*	0.358	−0.380	1.644	* 1.592	1.114	0.0019	1
44	*Anopheles barberi*	−0.109	−0.028	1.019	* 2.533	1.096	0.0001	1
45	*Aedes dupreei*	0.188	−0.080	0.764	* 2.084	0.967	0.0002	1
46	*Coquillettidia perturbans*	0.963	−0.142	1.261	* 0.412	0.822	0.0004	1
47	*Culex erythrothorax*	−0.094	−0.166	0.715	* 1.944	0.811	0.0004	1
48	*Anopheles franciscanus*	0.455	−0.328	1.037	* 0.997	0.771	0.0014	1
49	*Culex territans*	0.322	−0.115	0.913	* 1.198	0.766	0.0003	1
50	*Aedes melanimon*	0.135	−0.080	1.302	* 0.549	0.612	0.0002	1
51	*Culiseta incidens*	−0.529	−0.267	0.596	* 1.722	0.556	0.001	1
52	*Culiseta melanura*	−0.193	−0.080	0.686	* 1.036	0.478	0.0002	1
53	*Anopheles freeborni*	−0.252	−0.115	0.379	* 1.288	0.446	0.0003	1
54	*Aedes vexans*	−0.458	−0.328	0.453	* 1.388	0.425	0.0014	1
55	*Aedes dorsalis*	−0.109	−0.028	* 0.207	* 0.889	0.315	0.0001	2
56	*Psorophora columbiae*	−0.476	−0.305	0.516	* 0.958	0.299	0.0013	1
57	*Psorophora signipennis*	−0.554	−0.280	0.663	* 0.556	0.188	0.0011	1
58	*Orthopodomyia alba*	−0.109	−0.028	0.267	* 0.294	0.140	0.0001	1
59	*Culiseta impatiens*	0.500	−0.115	0.159	* −0.234	0.129	0.0003	1
60	*Aedes nigromaculis*	−0.109	−0.028	−0.151	* −0.093	−0.113	0.0001	1

* Indicates that the mosquito species has been reported as positive for the associated MBFV.

**Table 2 insects-12-00398-t002:** Regression coefficient and corresponding *t*- and *p*-values for each model.

Model	*R^2^*	*F*	*p*	*t*	*p*
DENV	0.71	19.63	0.002	4.43	0.002
YFV	0.46	6.81	0.031	2.61	0.031
SLEV	0.72	20.58	0.002	4.54	0.002
WNV	0.0003	0.00	0.962	−0.05	0.962
Weighted epsilon	0.64	14.49	0.005	3.81	0.005
Sample area	0.91	82.04	0.0002	9.06	0.0002

## Data Availability

All the Data used in this study can be accessed as shown in references [35,36,37,38,39,40].

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
