# Peer review of "Using Data Mining and Network Analysis to Infer Arboviral Dynamics: The Case of Mosquito-Borne Flaviviruses Reported in Mexico"

_insects, 2021, doi:10.3390/insects12050398_

Round 1

Reviewer 1 Report

This study uses a novel and interesting modelling framework to predict the importance and distribution of different mosquito species as vectors of medically important mosquito-borne flaviviruses in Mexico. Importantly the authors incorporated potential zoonotic reservoirs into the models and, based on the associations of mosquitoes and mammalian hosts, identify additional mosquito species that may serve as vectors for multiple MBFVs. Both the methodology and the results will be of interest to those involved in the study and surveillance of mosquitoes and mosquito borne diseases. This approach could be applied to other vector borne disease systems and help direct surveillance activities that monitor for VBDs or aim to identify potential vectors, so will likely also be of interest to a wider audience interested in ecology of zoonotic disease and enzootic cycles of VBDs.

The paper is very well written and the complex methodology is clearly explained and justified at each step. Results are interesting and well-presented. Overall I think this is an excellent study that deserves to be published. There are some areas where the clarity and scientific soundness could be improved. See detailed comments below:

Please check the species names throughout as there are many instances where they are not italicised. 

line229: or the mosquito may not preferentially feed on that particular mammal.

line 278-280: a major flaw with this sentence is that both WNV and SLEV are maintained in birds, although may be transmitted to mammals via mosquitoes. Mammals are not competent hosts for these 2 viruses, which is suggested by this sentence. However, this does not mean that the approach is not valid, as the presence of WNV- or SLEV-infected mammals suggests WNV is present in birds in that area from where it can spillover and infect mammals.
So the authors need to explain that this is the case to prevent misunderstanding by readers.
Similarly, in the next paragraph the infected mammal species are referred to as "confirmed" or "known" mammal hosts. As these mammals have only been confirmed positive for the viruses, it does not make them known reservoir hosts (which is implied, even if not intended, by the use of the word host), it merely shows that they can be infected. Therefore to avoid confusion, references to confirmed or known "hosts" throughout the manuscript should be renamed as "infected mammal species", etc. or "mammals confirmed positive" as is used later in the paper.

Figure 1 and 4. To improve presentation, the species names should either be all above the bars or all within. It is unclear why on the left half the graph has names above bars and the right half has names within bars.

Figure 3. Whilst the figure is lovely, some of the text labels in the figure are very small, I had to zoom in quite far to be able to read them. You may need to increase the size/orientation of the figure to improve visibility, as the text size cannot really be increased without obscuring other parts of the figure - perhaps an editorial decision?

Figure 5 & 6. Here it seems that the paper is missing the next logical step and these predictive maps could be quite easily validated by comparing them to the known distribution of these mosquito species and human cases of the MBDs from surveillance data. Whilst I understand this data might not be easily accessible and I do not know how good surveillance is for mosquitoes & MBDs in Mexico, the authors could discuss this in more detail in the Discussion if it is not possible to achieve for the results.

line 536-539: please check this sentence, it does not completely make sense. Perhaps some typos and/or punctuation missing.

line 569: perhaps this paragraph could also include some discussion on the possibility of using avian hosts in the model, as important hosts for WNV and SLEV maintenance.

Author Response

We thank the referee for their detailed report, positive comments and relevant criticisms. We address their detailed comments below:

Point 1.

line229: or the mosquito may not preferentially feed on that particular mammal.

*** Agreed. We have changed this.

Point 2.

line 278-280: a major flaw with this sentence is that both WNV and SLEV are maintained in birds, although may be transmitted to mammals via mosquitoes. Mammals are not competent hosts for these 2 viruses, which is suggested by this sentence. However, this does not mean that the approach is not valid, as the presence of WNV- or SLEV-infected mammals suggests WNV is present in birds in that area from where it can spillover and infect mammals. So the authors need to explain that this is the case to prevent misunderstanding by readers. Similarly, in the next paragraph the infected mammal species are referred to as "confirmed" or "known" mammal hosts. As these mammals have only been confirmed positive for the viruses, it does not make them known reservoir hosts (which is implied, even if not intended, by the use of the word host), it merely shows that they can be infected. Therefore to avoid confusion, references to confirmed or known "hosts" throughout the manuscript should be renamed as "infected mammal species", etc. or "mammals confirmed positive" as is used later in the paper.

*** We understand completely the referee’s point here. In general, it would be preferable to include in ALL potential biotic factors, not just mammals, or even just mammals and birds. Here, we wanted to show the feasibility of using our methodology to better understand the transmission cycle of MBFVs using the best data that was available in Mexico. For the chosen MBFVs this meant using mammals that had been identified as positive as birds have not been identified as positive for DENV or Yellow fever in Mexico, meaning that our multi-pathogenic viewpoint would be much weakened, whereas mammals have for all considered MBFVs. Of course, the existence of multiple biotic factors can lead to confounding, whereby a biota-mosquito co-occurrence, such as a mammal, is confounded by another biota, such as a bird. One of the main reasons we have not considered potential avian hosts in the paper is that there is much less information about them in Mexico. Additionally, although there is ample information outside of Mexico for WNV and SLEV that does not help in a co-occurrence analysis based on Mexican data. We will certainly return to this point in a future paper however, as our overall methodology allows for the identification of confounders so as to, for example, better understand if the direct biotic component is a bird or a mammal. We have added in some text to emphasize the referee's comment on this point.   

We have also changed references to confirmed or known hosts to the less charged “mammals confirmed positive” throughout the manuscript.

Point 3.

Figure 1 and 4. To improve presentation, the species names should either be all above the bars or all within. It is unclear why on the left half the graph has names above bars and the right half has names within bars.

*** We have now provided new graphs

Point 4.

Figure 3. Whilst the figure is lovely, some of the text labels in the figure are very small, I had to zoom in quite far to be able to read them. You may need to increase the size/orientation of the figure to improve visibility, as the text size cannot really be increased without obscuring other parts of the figure - perhaps an editorial decision?

*** We have provided the original .tif file which is much higher resolution. We believe this is better than dividing the figure into 4 separate ones.

Point 5.

Figure 5 & 6. Here it seems that the paper is missing the next logical step and these predictive maps could be quite easily validated by comparing them to the known distribution of these mosquito species and human cases of the MBDs from surveillance data. Whilst I understand this data might not be easily accessible and I do not know how good surveillance is for mosquitoes & MBDs in Mexico, the authors could discuss this in more detail in the Discussion if it is not possible to achieve for the results.

*** Once again, the chief barrier here is a data one. We do not have access to detailed case data. DENV would be a possibility as reporting it is obligatory but YFV - no cases - and WNV and SLEV very little. Additionally from the data we have seen in an "informal" setting, there are doubts about its validity that are probably a reflection of vagaries in the Mexican health system. Additionally, We have added in some discussion of these points in the Discussion section. With respect to the mosquito distributions, there is a potential for comparison there. However, this would require a substantial amount of extra analysis where we would have to go back and divide our collection data into training and test sets etc. We would also need to compare various benchmarks to compare model performance - climatic variables alone, all mammals independently of if or not they have been confirmed positive etc. Of course, we do not wish to state that this is not a worthwhile goal but, rather, wish to emphasize that the chief predictive element of the current analysis is the identification and ranking of the chief agents involved in the transmission cycle and identify potential new agents.

Point 6.

line 536-539: please check this sentence, it does not completely make sense. Perhaps some typos and/or punctuation missing.

*** Agreed. We have changed it.

Point 7. 

line 569: perhaps this paragraph could also include some discussion on the possibility of using avian hosts in the model, as important hosts for WNV and SLEV maintenance.

*** We have added in a paragraph in the Discussion section in reference to this point.

Reviewer 2 Report

“Using data mining and network analysis to infer arboviral dynamics: The case of mosquito-borne flaviviruses reported in Mexico” by Sotomayor-Bonilla et al.

This article aims to use data mining in order to predict the possible arboviral transmission of multiple pathogens between suspected mosquito vectors and possible mammalian reservoirs.

 While many studies analyzed the interactions between a single pathogen- a single vector and a single host the authors in the current study have a more environmental approach and challenged their model with populations dynamics of different viruses, different vectors and different mammalian reservoirs.

The approach presented in this article is novel. I hope to read more studies in the field of arboviral transmission with such a wide environmental approach.

However, this study could improve with several changes:

  • While the manuscript itself is well written and fairly clear, the abstract and summary should be rewritten to better present the study, its objective and results.
  • The authors did not look at the transmission of two globally very important flaviviruses: Zika and Chikungunya. Why? I think these viruses should be included in the analysis and if not – it should be clearly explained why the authors choose to ignore them.
  • The authors only examined possible mammalian reservoirs. WNV for example, as well as other flaviviruses, are known to have avian reservoirs. I think that the model should have also included avian reservoirs and specifically migratory birds. If the authors could not include avian reservoirs in their model- such a choice should be well explained and discussed.

Author Response

We thank the referee for their overall very positive comments and address their specific criticisms below: 

Point 1. 

  • While the manuscript itself is well written and fairly clear, the abstract and summary should be rewritten to better present the study, its objective and results.

*** We have tried to rewrite the summary and abstract. We hope this suffices. Without more specific information about which parts did not represent the study, its objectives and results well in the referee's opinion we have had to guess. We apologize if it should have been obvious where the weakness are. Sometimes it is difficult to step back from what one has written.

Point 2.

  • The authors did not look at the transmission of two globally very important flaviviruses: Zika and Chikungunya. Why? I think these viruses should be included in the analysis and if not – it should be clearly explained why the authors choose to ignore them.

*** We have added explanation as to this point. Of course, these MBFV are very important. However, the method we have adopted in this paper is to model the potential distribution of vectors using information about mammals  that have been confirmed as positive for the distinct MBFVs. We also require that the information be such that the collection data for both mosquitos and potential hosts (meaning species that have been found to be positive for some MBFV and species that in particular that have been found to be positive for ZIKV and/or CHIKV) is in the same place so as to permit a co-occurrence analysis. In the case of ZIKV and CHIKV we simply do not have that data in hand. Although we can build a model for a given vector for ZIKV or CHIKV using completely unknown potential mammal hosts that would be outside the methodology of this paper although this was done in a related paper - González-Salazar C, Stephens C, Sánchez-Cordero V. Predicting the Potential Role of Non-human Hosts in Zika Virus Maintenance. Ecohealth. 2017; 14(1):171–7.

Point 3.

  • The authors only examined possible mammalian reservoirs. WNV for example, as well as other flaviviruses, are known to have avian reservoirs. I think that the model should have also included avian reservoirs and specifically migratory birds. If the authors could not include avian reservoirs in their model- such a choice should be well explained and discussed.

*** We have added an ample discussion of this point in the manuscript as the other referee had exactly the same comment. In general, it would be preferable to include in ALL potential biotic factors, not just mammals, or even just mammals and birds. Here, we wanted to show the feasibility of using our methodology to better understand the transmission cycle of MBFVs using the best data that was available in Mexico. For the chosen MBFVs this meant using mammals that had been identified as positive as birds have not been identified as positive for DENV or YFV in Mexico, meaning that our multi-pathogenic viewpoint would be much weakened, whereas mammals have for all considered MBFVs. Of course, the existence of multiple biotic factors can lead to confounding, whereby a biota-mosquito co-occurrence, such as a mammal, is confounded by another biota, such as a bird. One of the main reasons we have not considered potential avian hosts in the paper is that there is much less information about them in Mexico. Additionally, although there is ample information outside of Mexico for WNV and SLEV that does not help in a co-occurrence analysis based on Mexican data. We will certainly return to this point in a future paper however, as our overall methodology allows for the identification of confounders so as to, for example, better understand if the direct biotic component is more likely to be a bird or a mammal.